# UV/Ozone-Treated and Sol–Gel-Processed Y_2_O_3_ Insulators Prepared Using Gelation-Delaying Precursors

**DOI:** 10.3390/nano14090791

**Published:** 2024-05-01

**Authors:** Sangwoo Lee, Yoonjin Cho, Seongwon Heo, Jin-Hyuk Bae, In-Man Kang, Kwangeun Kim, Won-Yong Lee, Jaewon Jang

**Affiliations:** 1School of Electronic and Electrical Engineering, Kyungpook National University, Daegu 41566, Republic of Korea; sangw98@knu.ac.kr (S.L.); pos03034@knu.ac.kr (S.H.); jhbae@ee.knu.ac.kr (J.-H.B.); imkang@ee.knu.ac.kr (I.-M.K.); 2School of Electronics Engineering, Kyungpook National University, Daegu 41566, Republic of Korea; 3School of Electronics and Information Engineering, Korea Aerospace University, Goyang 10540, Republic of Korea; kke@kau.ac.kr; 4The Institute of Electronic Technology, Kyungpook National University, Daegu 41566, Republic of Korea

**Keywords:** sol–gel method, Y_2_O_3_, RRAM, gelation, UV/ozone

## Abstract

In this study, a Y_2_O_3_ insulator was fabricated via the sol–gel process and the effect of precursors and annealing processes on its electrical performance was studied. Yttrium(III) acetate hydrate, yttrium(III) nitrate tetrahydrate, yttrium isopropoxide oxide, and yttrium(III) tris (isopropoxide) were used as precursors, and UV/ozone treatment and high-temperature annealing were performed to obtain Y_2_O_3_ films from the precursors. The structure and surface morphologies of the films were characterized via grazing-incidence X-ray diffraction and scanning probe microscopy. Chemical component analysis was performed via X-ray spectroscopy. Electrical insulator characteristics were analyzed based on current density versus electrical field data and frequency-dependent dielectric constants. The Y_2_O_3_ films fabricated using the acetate precursor and subjected to the UV/ozone treatment showed a uniform and flat surface morphology with the lowest number of oxygen vacancy defects and unwanted byproducts. The corresponding fabricated capacitors showed the lowest current density (*J_g_*) value of 10^−8^ A/cm^2^ at 1 MV/cm and a stable dielectric constant in a frequency range of 20 Hz–100 KHz. At 20 Hz, the dielectric constant was 12.28, which decreased to 10.5 at 10^5^ Hz. The results indicate that high-quality, high-k insulators can be fabricated for flexible electronics using suitable precursors and the suggested low-temperature fabrication methods.

## 1. Introduction

Flexible electronics overcome the mechanical issues of conventional rigid electronics; therefore, they are crucial for next-generation devices. They undergo deformation and are thin, economical, and bio-implantable. Therefore, they have wide applicability, such as in textiles, wearable sensors used to monitor health and detect bio-signals, rollable or flexible displays, and flexible solar cells and piezoelectric energy harvesting systems used to efficiently harvest energy [1,2,3,4,5,6].

These devices are fabricated using functional materials such as conductive polymers, organic semiconductor, and amorphous silicon (Si) [5,6]. However, the thermal and chemical instabilities of conductive polymers and organic semiconductors are unsuitable for fabricating complementary metal-oxide semiconductor (CMOS) devices. Devices fabricated using amorphous Si show poor performance. Therefore, metal oxides with higher mobility, transmittance, thermal stability, and process compatibility are used for fabricating CMOS devices. Metal oxides can be used to fabricate various devices such as conductors, semiconductors, and insulators [7,8,9,10,11,12,13]. Dielectric materials are used as electrical insulators as they are poor conductors of electricity. They are critical components in electronic devices, including transistors, resistive switching memory devices, ferroelectric memory devices, neuromorphic devices, and conventional data-storage devices such as flash memory devices [9,10,11,14,15,16,17].

High-quality functional metal-oxide devices, such as electrodes, semiconductors, and insulators, are fabricated by the sol–gel process using a liquid-phase solution. This prepared liquid-phase ink is used for spin coating, screening, and printing for large-area applications at low cost and shorter processing times, without requiring high vacuum conditions [18,19]. However, high-temperature annealing must be performed to obtain metal oxides from precursors at annealing temperatures higher than the deformation temperature of plastic substrates [14,15,16,17,18,19]. Thus, depositing high-quality metal oxides on plastic substrates for flexible electronics is difficult.

Herein, Y_2_O_3_ insulators were fabricated via the sol–gel process assisted by the ultraviolet (UV)/ozone treatment instead of high-temperature annealing. The UV/ozone treatment successfully eliminated unwanted products and suppressed the formation of oxygen vacancies, which can cause large variations in dielectric constants. In the sol–gel system, the fast gelation leads to the formation of network chains and makes it hard to form the uniform and flat films. Among different used precursors, the acetate precursor delayed gelation and yielded a uniform and flat Y_2_O_3_ layer on the substrate. Indium tin oxide (ITO)/Y_2_O_3_/Au capacitors were also fabricated on glass substrates using the acetate precursor and UV/ozone-treatment-assisted photochemical annealing below 200 °C. These capacitors had the lowest current density (*J_g_*) of 10^−8^ A/cm^2^ at 1 MV/cm and stable dielectric constant at 20 Hz–100 KHz. At 20 Hz, the dielectric constant was 12.28, which decreased to 10.5 at 10^5^ Hz. These results indicate that high-quality, high-k insulators can be fabricated for flexible electronics using appropriate precursors and low-temperature fabrication methods.

## 2. Materials and Methods

Yttrium(III) acetate hydrate, yttrium(III) nitrate tetrahydrate, yttrium isopropoxide oxide, and yttrium(III) tris (isopropoxide) were used as precursors and purchased from Sigma-Aldrich. These precursors were separately dissolved in 5 mL of 2-methoxymethanol (Aldrich, St. Louis, MO, USA, 99.9%) with a concentration of 0.20 M. Then, 0.1 mL of ethanolamine was added as a stabilizer to each solution, followed by sonication for 10 min for complete mixing. The electrical characteristics of insulator layers were tested by fabricating ITO/Y_2_O_3_/Au capacitors. A glass substrate coated with ITO (Aldrich) was sonicated in acetone and deionized water for 10 min to remove any undesirable residues. The remaining impurities were removed using a nitrogen (N_2_) blower. The substrates were then optically cleaned for 1 h using an UV/ozone lamp to improve their hydrophilicity and remove organic pollutants.

The completely mixed Y_2_O_3_ solutions were then spin coated on the ITO surface at a rate of 500 rpm for 5 s and 3000 rpm for 90 s. Subsequently, the substrates were soft-dried on a hot plate at 120 °C for 10 min. To increase the film thickness and evaporate the solvent, the substrates were baked on a hot plate at 200 °C for 1 min, and the coating process was repeated. Each film was annealed under three conditions: UV/ozone irradiation for 3 h at 25 °C (wavelength: 254 nm; photon flux density: 16 mW/cm^2^; SEN Lights SSP16-110), annealing at 400 °C for 1 h, and annealing at 500 °C for 1 h on a hot plate in air. Finally, a 50 nm thick Au layer was deposited via thermal evaporation onto the annealed films and used as the electrode. Hereinafter, yttrium(III) acetate hydrate and yttrium(III) nitrate tetrahydrate are referred to as acetate and nitrate, respectively. The Y_2_O_3_ films prepared from acetate and nitrate and annealed under UV/ozone irradiation or at 400 °C and 500 °C are referred to as acetate-UV, nitrate-UV, acetate-400, and acetate-500, respectively.

The structural characteristics and crystallinity of the films were analyzed via grazing-incidence X-ray diffraction (GIXRD; X’pert Pro, PANalytical, Almelo, The Netherlands) with an incidence angle of 0.3° and a Cu Kα radiation wavelength of 1.54 Å. Their surface morphologies were analyzed via scanning probe microscopy (SPM; NX20, Park Systems, Santa Clara, CA, USA) and thicknesses and surfaces were analyzed via field-emission scanning electron microscopy (FE-SEM; SU8220, Hitachi, Tokyo, Japan). X-ray photoelectron spectroscopy (XPS; Nexsa, ThermoFisher, Waltham, MA, USA) was used to examine the chemical properties of the films. Their thermal properties were assessed via thermo-gravimetric analysis–differential scanning calorimetry (TGA–DSC; Discovery SDT 650, TA Instruments, New Castle, DE, USA) and optical characteristics were studied via UV–visible (UV–vis) spectroscopy (LAMBDA 265, Waltham, MA, USA). A semiconductor parameter analyzer (Keithley 2636B, Keithley Instruments, Cleveland, OH, USA) and probe station (MST T-4000A, Hwaseong, Republic of Korea) were used to analyze the electric properties of the films. Their dielectric constants were measured using a precision LCR meter (E4980A, Keysight, Santa Rosa, CA, USA).

## 3. Results

Figure 1 shows the solutions containing each precursor. Ethanolamine was added to all the solutions, followed by sonication for 10 min. Acetate and nitrate dissolved completely and appeared transparent, whereas yttrium(III) tris (isopropoxide) and yttrium isopropoxide oxide did not dissolve completely. In particular, yttrium isopropoxide oxide exhibited the least dissolution. Clean and uniform films were not formed when using yttrium(III) tris (isopropoxide) and yttrium isopropoxide oxide as the precursors. Therefore, the precursor solutions containing acetate and nitrate in 2-methoxymethanol were used to fabricate Y_2_O_3_ films.

Figure 2 shows the GIXRD patterns of the Y_2_O_3_ films prepared using the yttrium(III) acetate hydrate and yttrium(III) nitrate tetrahydrate precursors and annealed under the three conditions. No diffraction peaks were observed in the GIXRD patterns of the Y_2_O_3_ films even after annealing at 500 °C for 1 h, indicating that the films were amorphous.

Figure 3 shows the SEM images of the films, from which the effect of the precursor and annealing conditions on the morphology of the films can be deduced. The surface roughness considerably varied depending on the precursor used. The Y_2_O_3_ films prepared using the acetate precursor were flat and uniform, whereas those prepared using the nitrate precursor showed a rough, nanoripple-embedded surface morphology. The physical properties of the solvent affected the surface morphology of the films. When a solvent with a high vapor pressure and low surface tension is mixed with another solvent with a low vapor pressure and high surface tension in a co-solvent system, the former evaporates and the latter exhibits natural convection, thereby elevating the film surface [20,21,22,23]. However, the same capping materials and solvents were used in our study. Therefore, the aforementioned co-solvent theory cannot support the obtained result. Both the precursors formed Y(OH)_3_ by reacting with H_2_O, and the formed Y(OH)_3_ decomposed into Y_2_O_3_ under hydrothermal conditions. In the sol–gel process, high water content accelerates gelation. The nanoripple-embedded surface originated due to the formation of a precursor network during hydrolysis. Hydrolysis occurred immediately after spin-coating deposition of nitrate, which contained numerous water molecules, yielding an uneven surface morphology due to gelation and aggregation [24].

Figure 4 and Figure 5 show the XPS spectra of the Y_2_O_3_ films. All binding energy values were corrected to 284.5 eV based on the C1s peak. Figure 4a,b show the Y3d XPS spectra of the Y_2_O_3_ films deposited using acetate and nitrate precursors, respectively, as a function of annealing conditions. The Y3d spectra show two peaks at 157.1 and 159.2 eV, corresponding to Y 3d_5/2_ and Y 3d_3/2_, respectively. Both spectra show that under UV irradiation for 3 h, the binding energy consistently shifts toward lower values. The Y 3d_5/2_ and Y 3d_3/2_ peaks shifted from 157.1 to 155.6 eV and from 159.2 to 157.5 eV, respectively, indicating a reduction in the energy required to form Y–O bonds. Thus, Y–O bonds are easily formed under UV irradiation for 3 h. Figure 4c–h show the Y3d XPS spectra of the Y_2_O_3_ films, wherein two peaks corresponding to pure Y metal and Y–OH are observed at 155.98 and 157.99 eV, respectively, as well as at 158.8 (Y 3d_5/2_) and 160.6 eV (Y 3d_3/2_), respectively. The proportions of pure Y metal in acetate-UV and nitrate-UV were the lowest at 7% and 12%, respectively. Under other conditions, this value averaged to over 15%. The proportion of Y–OH was the lowest in acetate-UV and nitrate-UV (at 7%) and the highest (at 20%) in acetate-500, indicating a decrease in the number of Y–OH bonds and Y metal bands. Figure 5a–f show the O1s spectra of the Y_2_O_3_ films, with four peaks at 529.0, 531.4, 532.1, and 533.8 eV corresponding to the lattice oxygen (O_L_), oxygen vacancies (O_V_), hydroxyl group (–OH), and O=C–O, respectively. Figure 5a–c show the O1s spectra of the Y_2_O_3_ films prepared using the acetate precursor. The relative area percentages of each component to the total area are estimated. For the acetate-UV films, the proportion of O_L_ was 84%, while those of O_V_, –OH, and O=C–O peaks were 11%, 4%, and 1%, respectively. For acetate-400, the proportion of O_L_, O_V_, –OH, and O=C–O was 16%, 61%, 21%, and 2%, respectively. For acetate-500, the proportion of O_L_, O_V_, –OH, and O=C–O was 13%, 62%, 8%, and 17%, respectively. Thus, as the annealing temperature increased, the proportion of O_V_ increased and the proportion of O_L_ considerably decreased. Figure 5d–f show the O1s spectra of the films prepared using the nitrate precursor, which are similar to those of the films prepared using the acetate precursor. The proportion of O=C–O was <7% under all annealing conditions. For nitrate-UV, the proportion of O_L_ was the highest at 86%, while the proportion of O_V_ and –OH was 7% and 5%, respectively. As the annealing temperature increased, the proportion of O_L_ considerably decreased: 16% and 10% for nitrate-400 and nitrate-500, respectively. Concurrently, the proportion of O_V_ considerably increased: 39% and 61% for nitrate-400 and nitrate-500, respectively. In addition, the proportion of–OH also increased: 38% and 25% for nitrate-400 and nitrate-500, respectively. The UV/ozone treatment strengthened the Y–O bonds and passivated O_V_. Free oxygen atoms or radicals are generated under UV irradiation, which penetrate the deposited layer and repair O_V_ by occupying the sites [25]. Thus, the Y_2_O_3_ films deposited using the UV/ozone treatment showed the lowest O_V_ concentrations compared to those deposited under other conditions. The chemical composition of the Y_2_O_3_ films corresponding to different annealing conditions is shown in Figure 6. In the GIXRD data, obvious XRD peaks corresponding to Y_2_O_3_ were not observed. However, as confirmed from the XPS data, the prepared films are Y_2_O_3_ films with small amounts of byproducts such as Y–OH, Y, and O=C–O carbonates.

Figure 7a,b show the C1s and N1s XPS spectra of the Y_2_O_3_ films, respectively. The C1s peak at 285 eV originates from the carbon as a contaminant. In addition, a very weak signal at 289.3 eV originates from ethanolamine. The intensity of N1s peaks is relatively less compared to that of the peaks corresponding to other chemical components. The signal at 399.0 eV originates from ethanolamine, attaching to formed Y_2_O_3_ films [26].

Figure 8 shows the optical characteristics of the glass/Y_2_O_3_ films. Figure 8a,b show the transmittance spectra in the visible-light range from 400 to 750 nm of the Y_2_O_3_ films and bare glass substrates. The average transmittance of the glass/Y_2_O_3_ films was >80% in the visible-light range, similar to that of the glass substrate. This high transparency signifies that the Y_2_O_3_ films can be effectively deposited on transparent thin-film transistors due to their minimal absorption losses. Acetate-UV and nitrate-UV had transparency levels comparable to those of acetate-500 and nitrate-500, respectively. This indicates that high transparency can be achieved via UV annealing similar to high-temperature annealing. The optical bandgap of the Y_2_O_3_ films was determined using the linear section of the *(αhν)^2^* versus *hν* graph. The inset of Figure 8c,d shows the (αhv)^2^ versus *hν* graph based on the Tauc equation:(1)$${\left(\alpha h\nu \right)}^{n}=A(h\nu -E_{g})$$
where *α* is the absorption coefficient, *h* is Planck’s constant, ν is the frequency, and *E_g_* is the optical bandgap. The optical bandgap values of the films were ~4.1 eV, regardless of the fabrication conditions. This value is lower than that reported for cubic Y_2_O_3_ (5.6 eV). The defect sites on the surface can reduce the optical bandgap of the Y_2_O_3_ films [27].

The insulation characteristics of the Y_2_O_3_ films were investigated by fabricating a metal/insulator/metal capacitor, i.e., a ITO/Y_2_O_3_/Au structure. Figure 9a shows the characteristics of current density (*J_g_*) versus applied electric field (E) of the ITO/Y_2_O_3_/Au capacitor under different fabrication conditions. For an accurate comparison of *J_g_*, a thickness-normalized electric field (E) was used as the x-axis. Regardless of the precursors used, capacitors containing high-temperature-annealed Y_2_O_3_ films showed high *J_g_* values. In contrast, capacitors containing Y_2_O_3_ films fabricated via UV/ozone treatment showed lower *J_g_* values. Capacitors containing acetate-UV showed the lowest *J_g_* value of 10^−8^ A/cm^2^ at 1 MV/cm. The XPS spectra confirmed that the high-temperature-annealed Y_2_O_3_ films were rich in O_V_, which increased the leakage current density [28]. These vacancies generated deep-trap energy levels, activating mobile electrons. SEM images showed that Y_2_O_3_ films prepared using the nitrate precursor had a rough surface morphology due to shorter gelation time, which increased the local electrical field and leakage current [29].

Figure 9b shows the dielectric constant versus frequency plots of the ITO/Y_2_O_3_/Au capacitors under different fabrication conditions. The dielectric constant was calculated by measuring the capacitance and thickness of the film. All capacitors, except that based on acetate-UV, showed high dielectric constants at 20 Hz, which varied considerably across frequency ranges of 20 Hz–100 KHz. However, the capacitor based on acetate-UV showed a stable dielectric constant at 20 Hz–100 KHz. At 20 Hz, the dielectric constant was 12.28 and decreased to 10.5 at 10^5^ Hz.

The dielectric constants of high-temperature-annealed and acetate-based Y_2_O_3_ insulators considerably decreased with increasing frequency. High-temperature-annealed Y_2_O_3_ insulators showed high O_V_ concentrations and unwanted products, such as –OH and O=C–O, as shown in the XPS spectra. Trapped charges accumulated near interfaces in the devices, causing space-charge polarization [30]. O_V_, negative oxygen ions, and –OH groups generated dipole moments, causing polarization at the material interfaces under a biased external field. O_V_ functioned as a trap site that could interact with space-charge polarization. As the frequency increased, the triggered polarization could not keep up with the frequency changes. Thus, high dielectric constant values were observed below 1 KHz. Above 1 kHz, the orientation of dipoles influences the dielectric constant [31]. At such high frequencies, electron mobility interferes with the alignment of dipoles and disrupts charge accumulation within dielectric materials, decreasing the dielectric constant [32].

## 4. Conclusions

Herein, Y_2_O_3_ insulators were fabricated via the sol–gel process. Acetate and nitrate were used as precursors for depositing Y_2_O_3_ films. The latter showed a wavy and rougher surface morphology. Moreover, UV/ozone treatment and high-temperature annealing were also performed during film fabrication. Compared to Y_2_O_3_ films processed via high-temperature annealing, those subjected to the UV/ozone treatment showed enhanced Y–O binding and the lowest number of O_V_ defects with the lowest unwanted-byproduct concentration. The corresponding fabricated capacitors showed the lowest *J_g_* value of 10^−8^ A/cm^2^ at 1 MV/cm and a stable dielectric constant in a frequency range of 20 Hz–100 KHz. At 20 Hz, the dielectric constant was 12.28, which decreased to 10.5 at 10^5^ Hz. Thus, high-quality, high-k insulators for flexible electronics can be fabricated using suitable precursors and low-temperature fabrication methods.

## Figures and Tables

**Figure 1 nanomaterials-14-00791-f001:**
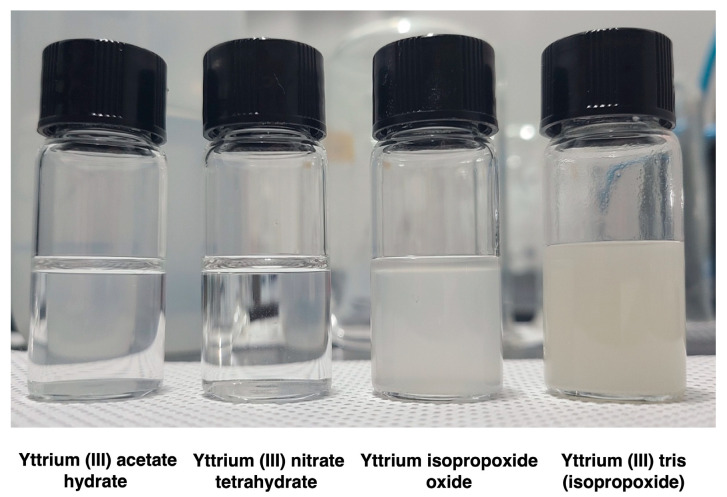
Photographs of the prepared solutions containing different precursors.

**Figure 2 nanomaterials-14-00791-f002:**
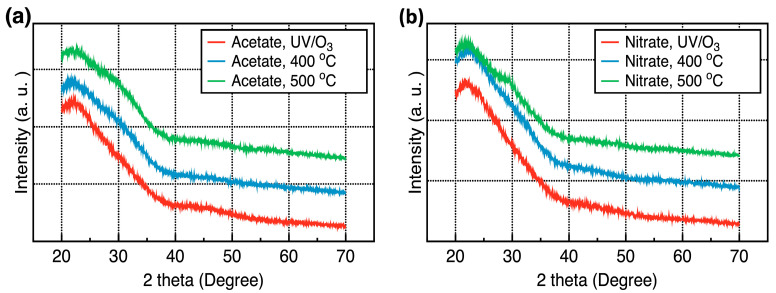
GIXRD patterns of the Y_2_O_3_ films corresponding to different annealing conditions: (**a**) acetate precursor; (**b**) nitrate precursor.

**Figure 3 nanomaterials-14-00791-f003:**
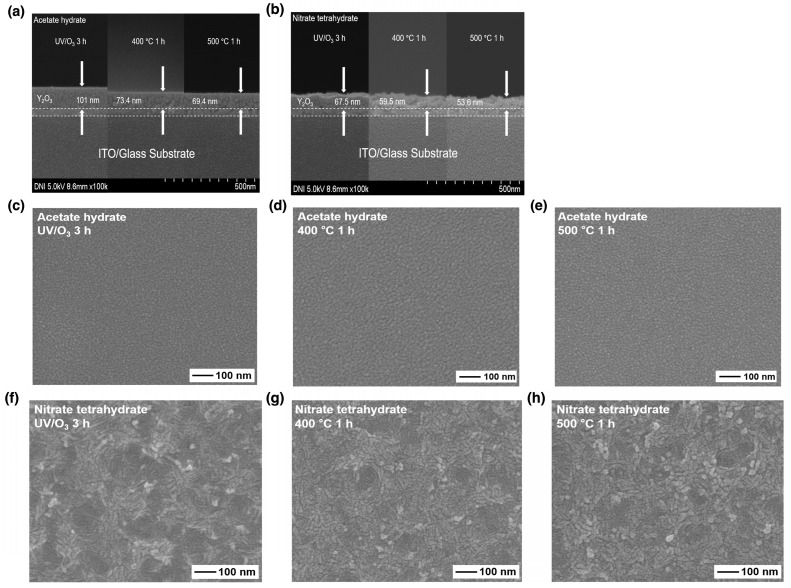
Cross-sectional SEM images of the Y_2_O_3_ films corresponding to different annealing conditions: (**a**) acetate and (**b**) nitrate precursor. (**c**–**h**) show the top-view SEM images.

**Figure 4 nanomaterials-14-00791-f004:**
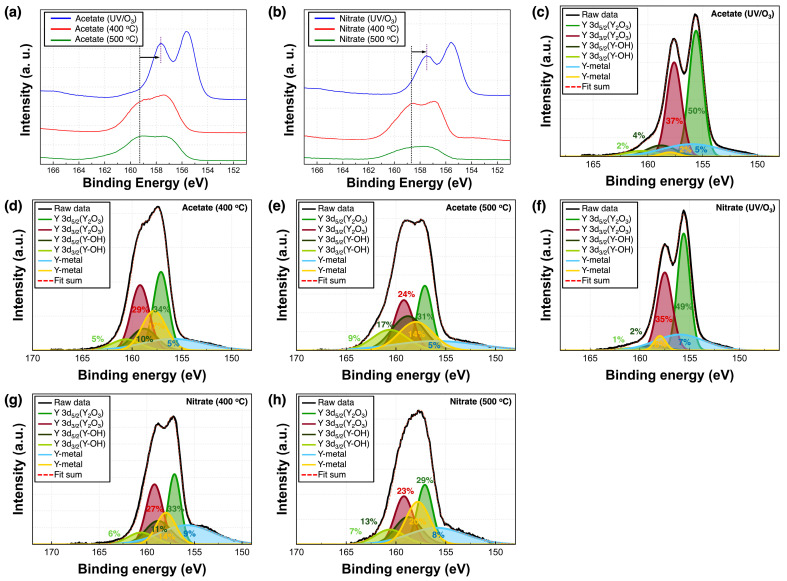
Y3d XPS spectra of the Y_2_O_3_ films corresponding to different annealing conditions: (**a**) acetate precursor; (**b**) nitrate precursor; (**c**–**h**) deconvoluted Y3d XPS spectra of the Y_2_O_3_ films.

**Figure 5 nanomaterials-14-00791-f005:**
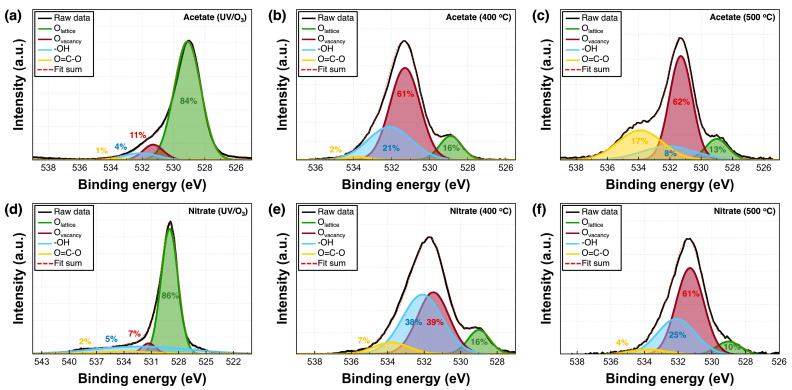
XPS O1s spectra of the Y_2_O_3_ films corresponding to different annealing conditions: Y_2_O_3_ films prepared using the (**a**–**c**) acetate and (**d**–**f**) nitrate precursors.

**Figure 6 nanomaterials-14-00791-f006:**
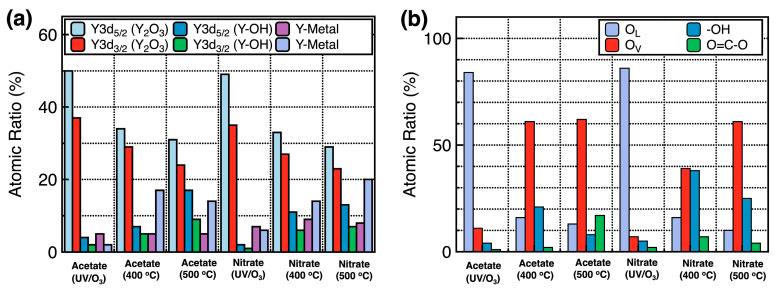
Compositional variation of the Y_2_O_3_ films corresponding to different annealing conditions, determined from Y3d XPS spectra (**a**) and O1s XPS spectra (**b**).

**Figure 7 nanomaterials-14-00791-f007:**
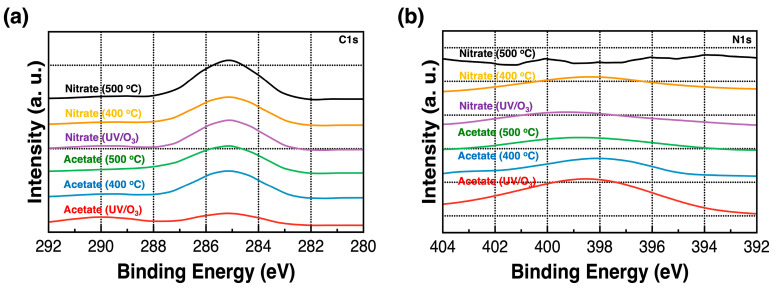
(**a**) C1s and (**b**) N1s XPS spectra of the Y_2_O_3_ films corresponding to different annealing conditions.

**Figure 8 nanomaterials-14-00791-f008:**
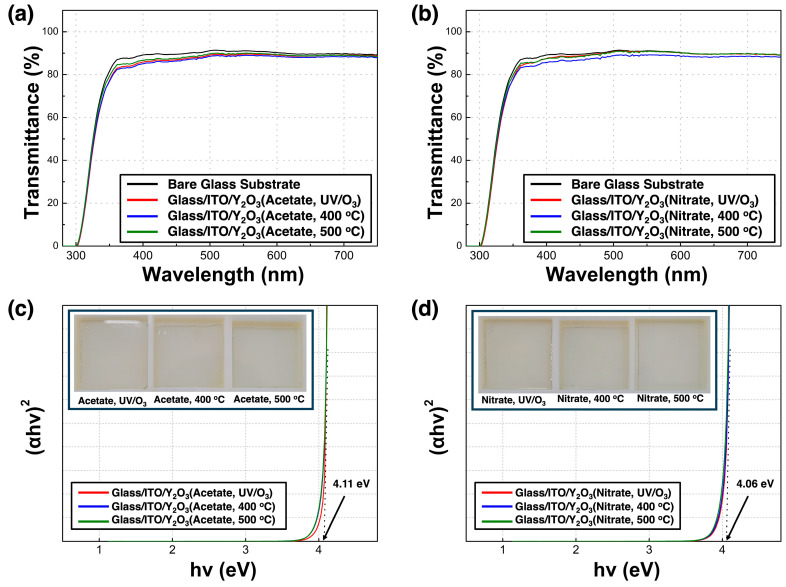
(**a**,**b**) Transmittance spectra of the glass/ITO/Y_2_O_3_ films; (**c**,**d**) (*αhν*)^2^ versus *hν* plot of the Y_2_O_3_ films corresponding to different annealing conditions. Inset shows the optical images of the films.

**Figure 9 nanomaterials-14-00791-f009:**
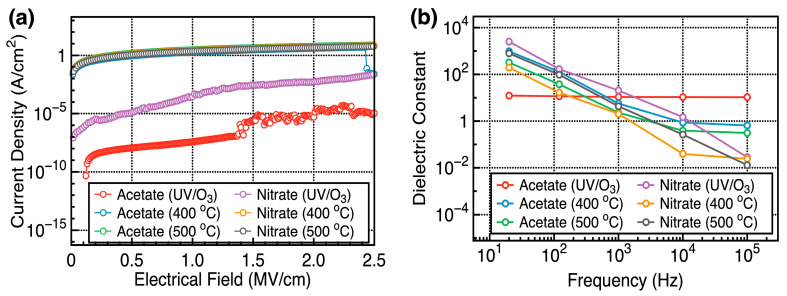
(**a**) Current density (*J_g_*) versus applied electric field (*E*) curves of the ITO/Y_2_O_3_/Au capacitors corresponding to different fabrication conditions. (**b**) Dielectric constant versus frequency curves of the ITO/Y_2_O_3_/Au capacitors corresponding to different conditions.

## Data Availability

Data are available in a publicly accessible repository.

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
