# Peer review of "UV/Ozone-Treated and Sol–Gel-Processed Y2O3 Insulators Prepared Using Gelation-Delaying Precursors"

_nanomaterials, 2024, doi:10.3390/nano14090791_

Round 1

Reviewer 1 Report (Previous Reviewer 2)

Comments and Suggestions for Authors

The authors addressed the comments.

Author Response

N/A

Reviewer 2 Report (New Reviewer)

Comments and Suggestions for Authors

Dear Authors,

you presented a well-designed and deep study about Yttrium oxide synthesis as a dielectric material for electronic devices.

Regarding the research design I have the follow concerns:

A) In the introduction is missing a comparison between electronic oxides dielectrics characteristics. It is just stated that :

"High-quality functional metal-oxide devices, such as electrodes, semiconductors, and 52 insulators, are fabricated by the sol–gel process using a liquid-phase solution. This pre- 53 pared liquid-phase ink is used for spin coating, screening, and printing for large-area ap- 54 plications at low cost and less processing times, not requiring high vacuum conditions [18, 55 19]. However, high-temperature annealing must be performed to obtain metal oxides 56 from precursors at annealing temperatures higher than the deformation temperature of 57 plastic substrates [14-19]. Thus, depositing high-quality metal oxides on plastic substrates 58 for flexible electronics is difficult"

This general sentence avoid to consider materials like ZnO that can be easily

grown on flexible substrates: 

https://doi.org/10.1016/j.materresbull.2017.03.030

and all the oxides that can be easily grown by physical deposition systems

like pulsed laser ablation or sputtering:

https://doi.org/10.1016/j.surfcoat.2015.06.052

B) the paper main result is about the good structural and electrical characteristics of UV-treated thin films, but only the one based on acetate precursor is behaving fine. Authors should discuss why nitrate yttrium oxide film is not behaving correctly since sctructural characterization is very similar.

C) Applied field is 1 V/cm not MV

D) Results on humid annealed films are scarcely of interest since it is well known that trapped humidity will escape from the sol-gel film damaging it

https://doi.org/10.1021/acs.chemrev.8b00045

E) Regarding electrical characterization it misses temperature/humidity stability of the dielectric behavior (possibly fig. 10) in a climatic chamber.

My best regards

Author Response

Reviewer 3 Report (New Reviewer)

Comments and Suggestions for Authors

This paper describes the electric properties of sol-gel-processed Y2O3 insulators prepared assisted by the UV/ozone treatment using gelation-delaying precursors. The results and discussions described in this paper are important for fabrication of flexible electronics. I recommend publication of this paper in Nanomaterials after the points descrived below are clarified and revised.

1) In page 2, line 89, it is worth to add the process temperature of UV/ozone trematment.

2) In page 4, figure 3, the intext process name "UV 3h" should be unified to such as "UV/O3 3h" in other figures.

3) In page 5, figure 4, there are unknown arrows in graphs (a) and (b), which need to be explaned.

4) In page 9, line 262, the dielectric constant of UV/O3 acetate remains of approximately the same order over a wide range of frequencies. The reason for this needs to be described.

5) IN page 9, figure 9, is it MV/cm instead of V/cm in the legend below, as shown in line 251 of Figure 9 on page 9?.

Round 2

Reviewer 2 Report (New Reviewer)

Comments and Suggestions for Authors

Dear authors,

I compliment for the detailed research,

still it is missing a very good introduction including

all the systems to produce flexible dielectrics,

neither you used flexible substrates.

Hopefully in future it will be done.

My best regards

Author Response

This manuscript is a resubmission of an earlier submission. The following is a list of the peer review reports and author responses from that submission.

Round 1

Reviewer 1 Report

Comments and Suggestions for Authors

The paper reports on the sol-gel assisted preparation of insulating yttrium oxide thin films. The subject of the paper is somewhat beyond the scope of Nanomaterials journal since no information on the nanostructures etc. is currently provided.

I have the following comments:

1. The paper is very hard to follow due to the poor English.

2. The Introduction section lacks information on the previous papers concerning sol-gel synthesis of yttrium oxide. More specifically, the same authors recently published another paper dealing with very similar synthetic approach, namely UV/ozone-assisted procedure for Y2O3 synthesis (doi 10.3390/ma15051899). Please provide the complete information on the background of the current research.

3. Please discuss the possible chemical interaction of the precursors, e.g. the interaction of yttrium salts with ethanolamine. Please provide the corresponding chemical reactions.

4. No adequate evidences are provided to confirm the formation of yttrium oxide. The films prepared are amorphous; no data are provided confirming the absence of carbon- or nitrogen-containing species in the annealed materials. Please provide information on the exact chemical composition of the films.

Comments on the Quality of English Language

Extensive editing of English language is required

Reviewer 2 Report

Comments and Suggestions for Authors

Overall comment:

Overall, the author prepared Y2O3 film on a glass substrate using different Yttrium precursors and different posttreatment methods such as UV/Ozone methods and high temperature annealing. The structure and morphologies are characterized using different techniques. The results show that Y2O3 films fabricated using the acetate precursor and UV/ozone process showed uniform and flat surface morphology with the lowest oxygen vacancy defects and unwanted byproducts.

Detailed comments are shown as following:

Comment 1: For Figure 2 description, detailed two precursors and three conditions need to be added from Line 118-121. Besides, the authors should explain in detail the GIXRD information that can be acquired. Why there is a up and down trend for the curve?

Comment 2: For Figure 3, the SEM images should be enlarged for a better view. Images in (c) look like covered by a grey filter which is really hard to see anything.

Comment 3:   The authors mentioned that fabricating flexible electronics in the introduction. However, in the main paper, the film is deposited on a glass substrate. The authors need to demonstrate the feasibility of soft substrate Y2O3 film formation.

Comment 4: For the XPS different chemical species summary in Figure 6, the doublet ratio for Y2O3 5/2 and 3/2 changes for different samples. This is not obeying the XPS analysis rules where the doublet should have a fixed ratio. (https://www.xpsfitting.com/2012/08/spin-orbit-splitting.html). Please check the XPS data analysis thoroughly and refer to the correct XPS analysis methods.

Comments on the Quality of English Language

English is fine.